# Discrimination and Characterization of the Volatile Organic Compounds in *Schizonepetae Spica* from Six Regions of China Using HS-GC-IMS and HS-SPME-GC-MS

**DOI:** 10.3390/molecules27144393

**Published:** 2022-07-08

**Authors:** Chao Li, Huiying Wan, Xinlong Wu, Jiaxin Yin, Limin Zhu, Hanjiang Chen, Xinbo Song, Lifeng Han, Wenzhi Yang, Heshui Yu, Zheng Li

**Affiliations:** 1College of Pharmaceutical Engineering of Traditional Chinese Medicine, Tianjin University of Traditional Chinese Medicine, Tianjin 301617, China; youchao_2020@163.com (C.L.); hywan1015@163.com (H.W.); 13261094746@163.com (X.W.); yiyi526117@163.com (J.Y.); songxinbo@tjutcm.edu.cn (X.S.); 2First Teaching Hospital of Tianjin University of Traditional Chinese Medicine, Tianjin 300193, China; zhulimin88120@163.com (L.Z.); chenhanjiang2012@163.com (H.C.); 3Haihe Laboratory of Modern Chinese Medicine, Tianjin University of Traditional Chinese Medicine, Tianjin 301617, China; hanlifeng_1@sohu.com (L.H.); ywz_0504@163.com (W.Y.); 4State Key Laboratory of Component-Based Chinese Medicine, Tianjin University of Traditional Chinese Medicine, Tianjin 301617, China

**Keywords:** volatile organic compounds, *Schizonepetae Spica*, HS-SPME-GC-MS, HS-GC-IMS, OPLS-DA

## Abstract

Volatile organic compounds (VOCs) are the main chemical components of *Schizonepetae Spica* (SS), which have positive effects on the quality evaluation of SS. In this study, HS-SPME-GC-MS (headspace solid-phase microextraction-gas chromatography-mass spectrometry) and HS-GC-IMS (headspace-gas chromatography-ion mobility spectrometry) were performed to characterize the VOCs of SS from six different regions. A total of 82 VOCs were identified. In addition, this work compared the suitability of two instruments to distinguish SS from different habitats. The regional classification using orthogonal partial least squares discriminant analysis (OPLS-DA) shows that the HS-GC-IMS method can classify samples better than the HS-SPME-GC-MS. This study provided a reference method for identification of the SS from different origins.

## 1. Introduction

*Schizonepetae Spica* (the dry spike of *Schizonepeta tenuifolia* Briq) is a traditional Chinese medicine (TCM). It distributes in Jiangsu, Zhejiang, Hebei, and Henan provinces in China. Clinical applications are widely used in colds, respiratory diseases, and skin diseases. Chemical studies showed that SS contained volatile organic compounds (VOCs), flavonoids, and organic acids, among which VOCs were the main medicinal component, and pharmacological activities of VOCs possess anti-inflammatory, antineoplastic, and antiviral properties [1,2,3,4,5,6,7]. Studies on SS mainly focus on volatile oil extraction and bioactive ingredients, but the overall characterization of VOCs of SS from different sources is not comprehensive enough [8].

At present, the most common used technologies for the comprehensive characterization of VOCs include gas chromatography-mass spectrometry (GC-MS), two-dimensional gas chromatography (GC×GC), gas chromatography olfactory determination mass spectrometry (GC-O-MS), and electronic nose (E-nose) [9]. GC-MS is being widely used in the analysis of VOCs, which has high-efficiency separation of gas chromatography and high resolution of mass spectrometry, but it has the drawback of cumbersome treatment before sampling and higher cost [10]. E-nose is a new aroma detection technology, which has the advantages of rapid detection, but it also has some problems, such as low precision, sensor drift, high sensitivity to the environment, and poor repeatability [11]. GC×GC is an analytical method with a powerful separation function, which is often used by the sample analysis of complex ingredients. However, GC×GC is not as widely used as GC-MS due to its high cost and complex experimental operation.

With the development of science and technology, more and more precision analytical instruments are used for the detection of VOCs, which provides more technical support for the research of TCM [12]. Headspace-gas chromatography-ion mobility spectrometry (HS-GC-IMS) is a new type of analysis method with the advantages of being fast, having high sensitivity, and no sample pre-treatment. In addition, the HS-GC-IMS has intuitive visualization of data, and is especially suitable for the variance analysis [13]. On the basis of GC-MS, headspace solid-phase microextraction-gas chromatography-mass spectrometry (HS-SPME-GC-MS) integrates the extraction and concentration technology of headspace solid phase microextraction and simplifies the sample pretreatment method [14]. At the same time, GC-MS has a high ability to isolate volatile compounds and provide detailed information of compounds [15]. Therefore, in this study, HS-GC-IMS and HS-SPME-GC-MS were used to comprehensively characterize the VOCs of SS. HS-GC-IMS usually detects small molecules of volatile components, which makes up for the deficiency of HS-SPME-GC-MS that can only capture medium molecules of volatile components [16]. The combination of these two analytical methods could easily, quickly, and accurately characterize the VOCs of SS.

It is well known that different production areas are often one of the important factors that led to the quality difference between the TCM. Hence, it is of great significance for the quality evaluation of TCM in different production regions. SS is one of the most popular traditional Chinese herbal medicines and distributed widely in China. In this study, SS from six regions were studied, including Hebei (B), Henan (N), Jiangsu (J), Shandong (S), Zhejiang (Z), and Anhui (A). In addition, a comprehensive strategy that integrated HS-SPME-GC-MS and HS-GC-IMS was proposed for the first time for rapid identification of chemical components of SS, with further treatment of data by different multivariate statistical analyses. In addition, this study also uses the same data processing program to compare two different hardware methods to distinguish the applicability of SS in different regions. Hopefully, by this example, new technical support can be provided for more convenient comparison of VOCs in different TCM.

## 2. Results

### 2.1. HS-GC-IMS Analysis

#### 2.1.1. The HS-GC-IMS Qualitative Analysis of VOCs

The VOCs of the SS in six different regions were analyzed by HS-GC-IMS. HS-GC-IMS depends on the fast ion-molecular reaction between air clusters and analytes generated by beta ionization and realizes the identification of VOCs [17]. We selected a batch of samples from six regions to make 2D top view, as shown in Figure 1A. The background color of the entire spectrum is blue, the abscissa represents the drift time, the ordinate represents the holding time, and the Ko is 2.032–2.034 cm^2^/Vs [18]. Different hues indicate different concentrations, with white dots indicating a lower concentration and red dots indicating a higher concentration, so the deeper the hue, the higher the concentration [19]. It was observed that the VOCs in SS from different regions were well separated at a retention time of 100–1000 s and drift time of 1.0–2.0 (Figure 1A). Based on the NIST library, VOCs of SS were determined by combining retention indexes (RI), retention times, drift times, and Ko [20]. These analysis results are shown in Table 1, in which forty compounds were tentatively identified. Forty distinct VOCs include five terpenoids (Figure 1B-a), eight alcohols (Figure 1B-b), ten aldehydes (Figure 1B-c), six esters (Figure 1B-d), five ketones (Figure 1B-e), two phenols, one acid, and three other substances.

#### 2.1.2. Differences in the Characteristic Volatile Fingerprints of SS

In order to more obviously compare the differences of samples from different regions, the chemical composition in the samples was classified and presented as a fingerprint in Figure 1B. The main VOCs in SS are terpenoid substances, as shown in Figure 1B-a. In each region, the highest concentration of substances is 1-menthol and d-camphor. Other terpenoids have a relatively high concentration in the two kinds of A and B. In general, A contains a high concentration of terpenoids. The alcohol substances in the SS samples are summarized in Figure 1B-b. The content of 2-hexanol has the highest content in the sample, but 3-heptanol and 3-furanmethanol have significant differences in the sample, and the content of these two substances is higher in the three kinds of J, A, and S. 1-Octen-3-ol content is generally low in Z. Interestingly, as can be seen from the Figure 1B, there are also differences among different batches within the same region, which may contribute to their different storage methods—for example, different storage times and temperature [21,22]. As shown in Figure 1B-c, the samples varied significantly according to the proportion of aldehyde substances in each. Trans-2-pentenal was higher in the J; (E)-hept-2-enal was higher in the N; benzaldehyde was higher in the J, A, and S. As shown in Figure 1B-d, there is no significant difference between samples according to the proportion of ester substances in each sample. There were relatively high levels of ethyl butanoate in J. The contents of ketone substances are summarized in Figure 1B-e. Acetophenone and cyclohexanone were present in high concentrations in all samples, but the 6-methyl-5-hepten-2-one had a relatively high concentration in B and nonan-2-one was higher in Z. In addition, there was a relatively high amount of p-cresol in A.

However, the limitations of the currently available library for HS-GC-IMS hinder the qualitative analysis of the VOCs of SS. As shown in Figure 1C, twenty-nine species were not identified. In these undefined compounds, Figure 1C-a summarize substances with high content in the sample. As shown in Figure 1C-b, the samples varied significantly in the undefined compounds.

### 2.2. Identification of VOCs by HS-SPME-GC-MS

In this study, forty-two VOCs were identified through HS-SPME-GC-MS, including nineteen terpenes, four alcohols, four esters, eleven ketones, one phenol, and three others, as summarized in Table 2. The results showed that l-menthone, (+)-pulegone, piperitone, menthofuran, verbenone, (+)-isomenthone, trans-carveyl acetate, and caryophyllene oxide were the main volatile compounds in all of the analyzed SS samples. Five substances passed the standard substance verification, as shown in Figure 2B, which was consistent with the identification results of the database. Surprisingly, cubebene was the volatile component only found in A and Z. Moreover, menthofuran was only found in the B and Z.

The MetaboAnalyst 5.0 was utilized for heat map clustering analysis to better understand the variations between SS samples in different regions. Each variable is normalized to generate the clustering heat map of SS in different regions on the basis of row-scale, using the relative contents of the discovered 42 compositions as variables. The distribution frequency of each substance in the sample is shown by row comparison. As shown in Figure 2C, the main VOCs in SS are terpenoids. (+)-pulegone is a key active component of SS essential oil and has been determined to have anti-inflammatory properties [23]. Compared with samples in other regions, those originating from A exhibit higher levels of pulegone. Moreover, the contents of other terpenoids (8–13) in A are significantly higher than those detected in other samples. As shown in Figure 2C, according to the Euclidean distance, samples from the six regions could be divided into three categories: B and N, Z, G and S, and A. The A has the highest content of terpenoids. Unlike the other four groups of samples, the samples from B and A are rich in ketones and esters. The contents of VOCs have a low concentration in the three kinds of J, Z, and S, respectively.

### 2.3. Comparison of the Recognition Abilities of HS-GC-IMS and HS-SPME-GC-MS for VOCs in SS in Different Regions

To further compare the ability of the two instruments to distinguish SS from different regions, the two sets of data were analyzed. In this study, OPLS-DA analysis was used to eliminate the interference of sample differences from different batches of the same Chinese medicinal materials on the VOC characteristics. OPLS-DA is usually applied as an approach to discriminate two or more groups and to model multiple classes simultaneously [24]. The characteristics of OPLS-DA are integrated orthogonal signal correction filters, which can separate system changes in the prediction (related to Y-related) and orthogonal (and Y-in-correlated) components. Therefore, OPLS-DA can reduce system noise and extract variable information, with stronger classification capabilities. He et al. used OPLS-DA to compare the applicability and predictive ability of E-nose and HS-SPME-GC-MS for the regional classification of Baijiu samples [25]. 

In our study, to achieve more accurate classification results, preprocessing is applied to experimental data. First, peak area collected from HS-SPME-GC-MS and HS-GC-IMS was subjected to a logarithmical transformation. The peak area acquired was transformed to log10, thus narrowing the scope of data [26]. It was then centered and pareto-scaled for the transformed variables. The Pareto scale, through dividing by the square root of the standard deviation of each column, for each variable provides a standard deviation that is equal to its initial variance [27]. For more than 50% of variables, the missing value has been excluded before the model is established, the datasets of different regional samples are introduced into the OPLS-DA model to perform region classification.

The OPLS-DA models are based on the results of HS-SPME-GC-MS and HS-GC-IMS; the whole sample size in the two models was set at 54 (six cultivars, three batches × triplicate). Seventy-four VOCs were used in the OPLS-DA model of HS-GC-IMS, while a total of 49 VOCs were used in the OPLS-DA model of HS-SPME-GC-MS. The sevenfold cross-validation procedures were used to validate the OPLS-DA models. The component is significant if the Y variation fraction was predicted by the X model >0.01 [25]. Then, based on analytical methods of both HS-GC-IMS and HS-SPME-GC-MS, five predictive components and eight orthogonal components were selected for the OPLS-DA model. As shown in Figure 3A,C, the classification model fitness of model based on HS-GC-IMS was 96.9% (R2Y(cum) = 0.969), while, for HS-SPME-GC-MS, it was 93.5% (Q2(cum) = 0.935). From the prediction ability point of view, the model built from HS-GC-IMS was as high as 95.5% (Q2(cum) = 0.935); however, it was only 85.9% (Q2(cum) = 0.859) correspondingly in HS-SPME-GC-MS. The higher values of R2 and Q2 indicated that the model based on HS-GC-IMS had better fitness and prediction ability compared with HS-SPME-GC-MS. From the predictive point of view, both models acquired Q2 > 0.5, indicating that there are good predictive capabilities, but the predictive ability of the HS-GC-IMS model is better. In order to better verify the ruggedness of the two OPLS-DA model, the two models have been exchanged for 200 replacement experiments. As shown in Figure 3B,D, the values of intercepts of randomly permutated models were significantly lower than that of the original one, indicating the ruggedness of the developed model [28].

As can be observed, in the score plot of OPLS-DA model of HS-GC-IMS, the SS from different regions show strong clustering, without any overlap. However, in the score plot of OPLS-DA mode of HS-SPME-GC-MS, SS samples of six geographical origins were not clearly discriminated. Samples from regions A, B, and C were not separated and samples from regions N and B also overlapped. This is similar to the result of heat map clustering in Figure 3. From this result, the difference between medium molecular weight VOCs of SS from different geographic is not obvious, but the compounds of small molecules identified by HS-GC-IMS have obtained better distinction. Therefore, we will further analyze the compounds identified by HS-GC-IMS.

### 2.4. Rapid Identification of SS in Different Regions by HS-GC-IMS

In general, due to the influence of growing environment, the composition and content of volatile components in the SS of different origins are also different. Therefore, it is particularly important to select SS from different regions. HS-GC-IMS takes less time to obtain analysis results, and data do not require complex processing. Differences between samples can be directly compared through the fingerprint generated by the machine. In this study, HS-GC-IMS was used for rapid identification of the volatile components of SS. The OPLS-DA model established based on the results of HS-GC-IMS had a high prediction ability for the panicles of SS from different habitats, and the samples obtained a good separation degree.

However, not all VOCs in various samples were significantly different. In order to see the difference, we analyzed the variable importance for the projection (VIP) predictive of VOCs in SS (Figure 4). VIP is generally used to evaluate the contributions of X-variables to a model [29]. Based on the criteria of VIP > 1, 30 (red) important variables were selected in the SS of different regions from the VIP plot of the OPLS-DA model of HS-GC-IMS, but there are unknown compounds in these variables, as shown in Figure 4A. In order to further explore advantageous biomarkers, a random forest model (RF) is established using known compounds. RF is an effective high-dimensional data analysis supervision method, which is a popular ensemble learning algorithm for classification and prediction [30]. The classification trees were set as 1000 in this study. During the construction of the tree, a third of the sample is excluded from the bootstrap sample (out-of-bag data, OOB data). For an unbiased estimation of the classification error (OOB error), the OOB data were used as test samples [31]. Academically, the lower the OOB error, the more accurate the classifier. In this experiment, after several trees, the cumulative OOB error rates dropped to 0.0185. As shown in Figure 4B, the 14 variables with significant differences contributed to the classification of SS in different regions. In A, p-cresol was more abundant. Trans-2-hexenal and 1-menthol are higher in B. J has greater amounts of ethyl acetate, 2-ethyl-5-methylpyrazine, and trans-2-pentenal. 1-phenylethanol was more abundant in the N. Diethyl trisulfide, 1-octen-3-ol, and 2-hexanol are higher in S. The levels of 3-methylbutanoic acid and alpha-phellandrene in Z are higher. These compounds are similar to the analysis results of Figure 1B. Those compounds with significant differences improve the accuracy of random forest classification, which may be a key factor in distinguishing from SS of different origins based on HS-GC-IMS.

## 3. Discussion

SS is one of the most important drugs for relieving exterior syndrome in TCM. The volatile oil distilled from SS showed potent anti-inflammatory and fumigant activity [32]. Steam distillation, solvent extraction, and headspace capture are commonly used to ex-tract volatile components from plants [33]. However, the first two methods require large sample size, long extraction time, and long heating time, and some components may be destroyed in the heating process. HS-SPME combined with GC-MS has the advantages of short extraction time, simple operation, and high sensitivity. At the same time, based on the NIST database search, this technology can realize the discovery of trace volatile components in complex sample systems [34]. HS-GC-IMS is a powerful technique for the separation and sensitive detection of VOCs, with the advantages of high sensitivity and resolution [35]. In the present study, the VOCs of SS were investigated by HS-GC-IMS and HS-SPME-GC-MS. As can be seen from Table 1 and Table 2, most volatile organics detected by HS-GC-IMS are small molecular compounds, whereas the detection range of HS-SPME-GC-MS is usually medium molecular weight VOCs. This suggests that the HS-GC-IMS has a higher sensitivity to high-volatile chemicals than the HS-SPME-GC-MS. It is the ability of SPME to capture high boiling point compounds that leads to the higher content of high boiling point compounds in volatile organic compounds detected by HS-SPME-GC-MS, which is consistent with previous reports [36]. In addition, SPME is affected by temperature, and the distribution coefficient decreases when the temperature increases. As a result, volatile compounds with low distribution coefficients cannot be detected by temperament. The combination of HS-GC-IMS and HS-SPME-GC-MS technology can better realize the rapid identification and comprehensive characterization of volatile organic compounds in SS. By using the well-established HS-GC-IMS technique, 40 VOCs were discovered in SS. On the other hand, 42 VOCs were identified in SS, by the established HS-SPME-GC-MS analysis method. The SS of volatile compounds may be affected by the cultivation region, resulting in the differentiation of the volatile composition in each SS sample. In this study, HS-SPME-GC-MS and HS-GC-IMS ware used to analyze the VOCs in 18 samples collected from different provinces in China. In total, terpenes ware the main volatile components in the SS. The results of both instruments indicated that there were more terpenoids in the samples from Anhui province. In addition, the two instruments reflect a certain difference in the content of volatile components of SS in different places, which may be related to factors such as climate conditions, soil conditions, sunshine intensity, cultivation conditions, and transportation conditions.

Modern studies show that there are obvious differences in the types and quantities of chemical components in TCM from different geographical sources [37]. Therefore, it is very important to establish a fast and reliable regional identification method for TCM. For this purpose, the analysis strategies of two different HS-SPME-GC-MS and HS-GC-IMS were selected, tested, and compared. In this study, the origin of SS samples from six regions was analyzed. A uniform statistical technique was used to analyze data collected on two analytical systems. Regional classification performed using OPLS-DA indicated that the method based on HS-GC-IMS could better classify the SS from six regions than HS-SPME-GC-MS. This may be because the environmental differences in different regions have a greater impact on the small molecule metabolites of plants. The characterization and classification of VOCs from Chinese herbal medicines by HS-GC-IMS coupled with an appropriate multivariate analysis has the potential to be used as a non-destructive way to evaluate Chinese herbal medicines from different origins. This gives us a new idea to distinguish Chinese herbal medicines in different regions. A number of studies have indicated that HS-GC-IMS has the capacity to confirm geographical and botanical origin [35]. For example, HS-GC-IMS was successfully used for reliable classification of geographical origins for both olive oil (EVOO) [38] and wine [39]. Furthermore, the HS-GC-IMS method was applied to quickly identify Ophiopogonis Radix from different regions [40]. Moreover, HS-GC-IMS has also been used to determine the geographical origins of “Chenpi” [41]. The compounds identified by HS-GC-IMS were subjected to multivariate analysis. A total of 14 volatiles that could explain the separation of SS samples into six regions were identified based on VIP scores and RF.

Present observations showed that HS-GC-IMS and HS-SPME-GC-MS can better characterize volatile components in TCM. The application of HS-GC-IMS can better distinguish between different sources of SS and improve the effectiveness and efficiency of the process. HS-GC-IMS has good potential in identifying TCM from different regions. However, the HS-GC-IMS database is not perfect, so it is still limited in the comprehensive characterization and accurate quantitative analysis of samples. Therefore, a large number of experiments are still needed to enrich the database for the application of HS-GC-IMS in distinguishing TCM from different origins.

## 4. Materials and Methods

### 4.1. Sample Source and Preparation

The SS samples were collected from different geographical localities, including Hebei, Henan, Jiangsu, Zhejiang, Shandong, and Anhui. Each sample bought three batches from one region, and there was no difference among batches. Detailed information of all the samples is listed in Appendix A. All samples in the experiments were authenticated by Professor Lijuan Zhang from Tianjin University of Traditional Chinese Medicine. Voucher specimens were deposited in the School of Pharmacy, Tianjin University of traditional Chinese medicine, China.

All SS samples were crushed with a grinder (Tai site, Tianjin, China) and sieved through a 40-mesh sieve. For subsequent examination, the powdered sample was immediately packed in a plastic bag and stored in a dark, dry environment of 20 °C.

### 4.2. Chemicals and Reagents

A C8-C20 n-alkane standard for HS-SPME-GC-MS was purchased from Sigma-Aldrich Trading Co., Ltd. (Shanghai, China). A C4-C9 n-ketones standard for HS-GC-IMS was purchased from Sinopharm Chemical Co., Ltd. (Shanghai, China). Reference compounds were purchased for identification. Limonene (CAS:5989-54-8, 95%), 3-methyl-Cyclohexanone (CAS:591-24-2, 97%) were bought from Aladdin Biochemical Technology Co., Ltd. (Shanghai, China). Piperitone (CAS:89-81-6, ≥95%), caryophyllene (CAS:87-44-5, ≥98%), caryophyleneoxide (1139-30-6, ≥90%) and (+)-pulegone (CAS:89-82-7, ≥98%) provided by Shanghai Yuanye Biotech. Co., Ltd. (Shanghai, China).

### 4.3. HS-GC-IMS Analysis Conditions

The HS-GC-IMS system (Flavourspec^®^, G.A.S, Dortmund, Germany) was equipped with an autosampler (CTC Analytics AG, Zwingen, Switzerland) and an FS-SE-54-CB-1 capillary column (15 m × 0.53 mm ID, 1 μm, CS-Chromatographie Service GmbH, Germany). A 0.2 g sample of SS was accurately weighed into a 20 mL headspace bottle and incubated at 75 °C for 20 min at 500 r/min. The volume of the extracted headspace air was 100 μL, and the syringe temperature was 45 °C. The column temperature was 60 °C, and the carrier gas consisted of 99.99% pure nitrogen and its flow rate was first set at 2 mL/min for 2 min, then increased to 10 mL/min within 15 min, increased to 100 mL/min over 25 min and increased to 120 mL/min over 30 min. The pre-separated compounds driven into an ionization chamber and ionized by a 3H ionization source with 300 MBq activity in positive ion mode. The resulting ions were driven to a drift tube (9.8 cm in length), which was operated on a constant temperature (45 °C) and voltage (5 kV). The flow rate of the drift gas (nitrogen gas) was set at 150 mL/min. The retention index (RI) of each compound was calculated using n-ketones C4-C9 as external references. VOCs were identified based on an IMS database of the HS-GC-IMS Library Search application software. The mobility Ko is also involved in the identification of compounds. It is a normalized expression of the ionic mobility (K) at standard temperature and pressure, and its calculation formula is based on references [42].

### 4.4. HS-SPME-GC-MS Analysis

#### 4.4.1. Extraction of Volatile Compounds

SPME fiber (Supelco, Inc., Bellefonte, PA, USA) was installed on a MultiPurpose sampler (Gerstel, GER) and combined with 7890 B–7000 D triple quadrupole gas chromatography mass spectrometry (Agilent Technologies, Palo Alto, CA, USA) to detect VOCs in SS samples. The univariate method was used to select the SPME conditions for each factor individually. Crushed samples (0.1 g) were placed into a 20 mL headspace vial with a magnetic screw cap and a Teflon-lined rubber septum. Then, the sample vial was equilibrated for 5 min at a certain temperature (50 °C, 60 °C, 70 °C, and 80 °C) on a heating platform. The extraction was conducted by inserting the SPME fibers (polyacrylate 85 µm, polydimethylsiloxane/divinylbenzene 65 µm phase thickness (PDMS/DVB), polydimethylsiloxane/carbon wide range/divinylbenzene 50/30 µm phase thickness (PDMS/CAR/DVB)) into the head space of the vial for a certain time (15 min, 20 min, 25 min and 30 min). The fiber was desorbed into the injection port of the GC for 5 min at the end of the extraction. Under the same conditions of sample size, extraction temperature and extraction time, the optimal SPME fiber was first determined. Then, the best SPME fiber was chosen to optimize the other factors at the same way, and peak capacity was used as the criterion. The resulted optimal extraction parameters were determined as follows, using PDMS/CAR/DVB fiber to extract 0.1 g of sample at 60 °C for 25 min.

#### 4.4.2. GC-MS Analysis

An Agilent 7890B gas chromatography system with an HP-5MS elastic quartz capillary column (30 m × 0.25 mm × 0.25 m, 19091 S–433, J&W Scientific, Folsom, CA, USA) and an Agilent 7000 D mass spectrometry detector was used for the GC-MS analysis. Helium was used as the carrier gas, with a flow rate of 1 mL/min. The heating procedure of the column was as follows: maintain a temperature of 50 °C for 2 min; increase to 70 °C at a rate of 10 °C /min; increase to 110 °C at a rate of 5 °C /min; increase to 115 °C at a rate of 1 °C /min; increase to 147.5 °C at a rate of 5 °C /min; increase to 160 °C at a rate of 3 °C /min; heated to 220 °C at 5 °C /min and held for 5 min. The mass spectrometry conditions were as follows: quadrupole temperature of 150 °C, ion source temperature of 230 °C, injection temperature of 280 °C, scanning range 50–600 *m*/*z*, and ionization voltage of 70 eV.

In this study, unknown volatile compounds were qualitatively analyzed by three methods, including searching for spectral peaks in the NIST17 standard mass spectrometry library of the chemistry workstation, calculating retention index (RI) values, and comparing the retention times of substances in the sample with external standards. The peak area normalization method was used to calculate the relative percentage content of each compound in SS samples from different regions.

### 4.5. Statistical Analysis

The SIMCA-P 14.1 software (Umerics, Umea, Sweden) was used to run the OPLS-DA. A Laboratory Analytical Viewer (LAV), three plugins, and a HS-GC-IMS library search are all included in the HS-GC-IMS assisted analysis software to analyze samples from different angles. The specific volatile compounds were identified by HS-GC-IMS library search software. Then, the galley plot program in LAV software was used to obtain the visual galley plot of the sample. The HS-SPME-GC-MS results were expressed as the mean ± standard deviation. The heat map and random forest ware performed by the online website MetaboAnalyst 5.0 for data processing.

## 5. Conclusions

In general, the study explored the 82 VOCs of SS from six regions by using HS-SPME-GC-MS and HS-GC-IMS. Among them, the content of terpenoids accounted for a large proportion of the SS. Through these two kinds of instrument analysis, it was concluded that the terpenoid substance content is higher in Anhui province. In addition, some aldehydes and phenols were found by HS-GC-IMS. The combination of the two analytical methods realized the rapid identification and comprehensive characterization of VOCs in SS. The fingerprint of HS-GC-IMS has the advantage of data visualization, which can directly and conveniently compare VOCs in samples from different regions. The discriminant analysis of SS from six regions was carried out by OPLS-DA. Only parts of SS from different regions can be separated by HS-SPME-GC-MS. HS-GC-IMS could effectively distinguish the six kinds of SS. By random forest analysis, 14 compounds were identified that were beneficial to the classification of SS in different areas. Although HS-GC-IMS was more commonly used to identify food from various sources, it was less commonly used in traditional Chinese medicine. HS-GC-IMS coupled with an appropriate multivariate analysis has potential in the characterization and classification of TCM containing VOCs. However, due to the complex composition of TCM, the application of HS-GC-IMS in the identification of TCM from different origins still needs to be studied with a larger sample size.

## Figures and Tables

**Figure 1 molecules-27-04393-f001:**
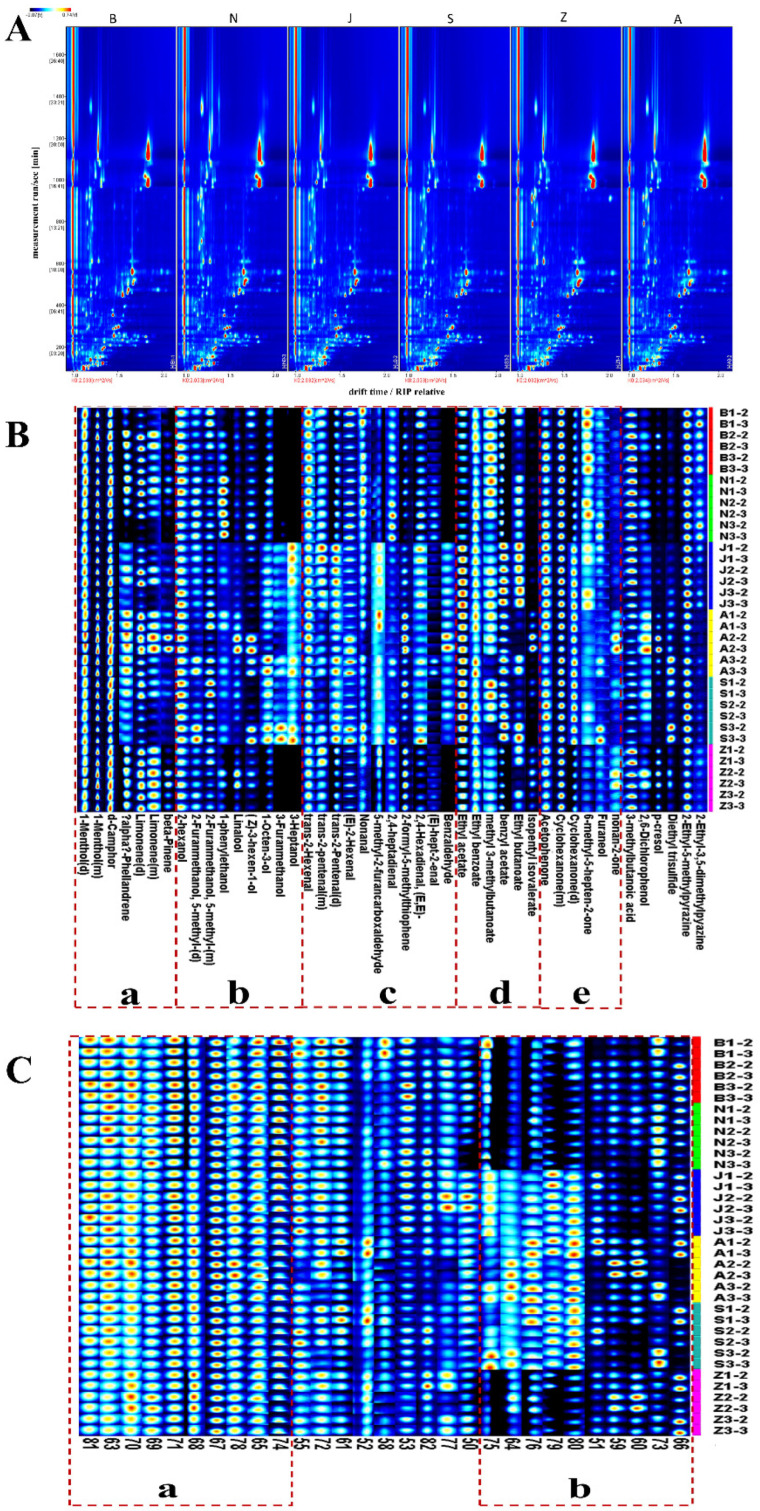
(**A**) Two-dimensional chromatogram results of VOCs in SS of different regions; (**B**) the VOCs fingerprint of different regions of SS; (**C**) the fingerprints of unknown compounds in SS from different regions.

**Figure 2 molecules-27-04393-f002:**
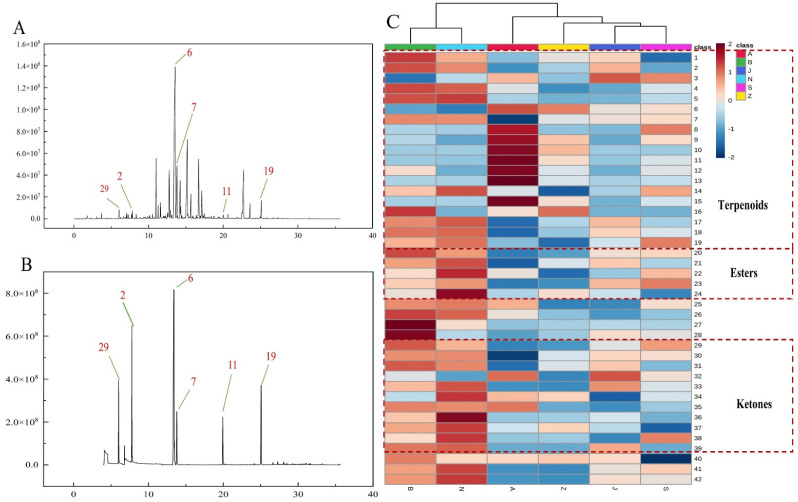
(**A**) Total ion chromatograms of volatile oils in SS of the B region; (**B**) reference compounds spectrogram; (**C**) the heat map clustering of the VOCs in SS samples in different regions (The codes of the compounds correspond to those in Table 2).

**Figure 3 molecules-27-04393-f003:**
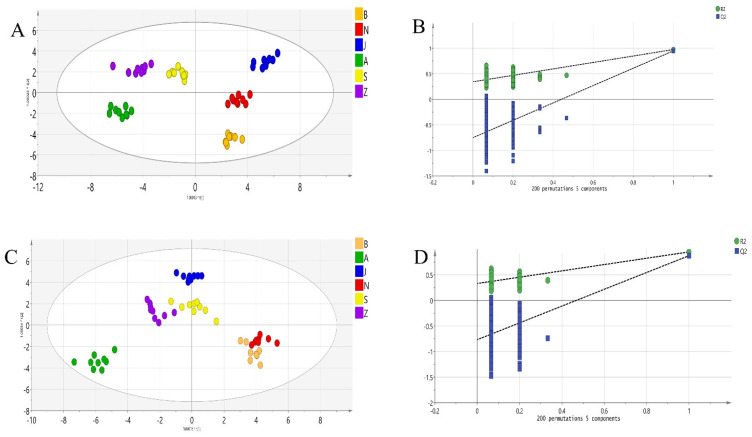
The OPLS-DA score plots of region multi-classification by HS-GC-IMS (**A**) and HS-SPME-GC-MS (**C**). (**B**) the permutation test results (*n* = 200) of the OPLS-DA model of HS-GC-IMS; (**D**) the per-mutation test results (*n* = 200) of OPLS-DA mode of HS-SPME-GC-MS.

**Figure 4 molecules-27-04393-f004:**
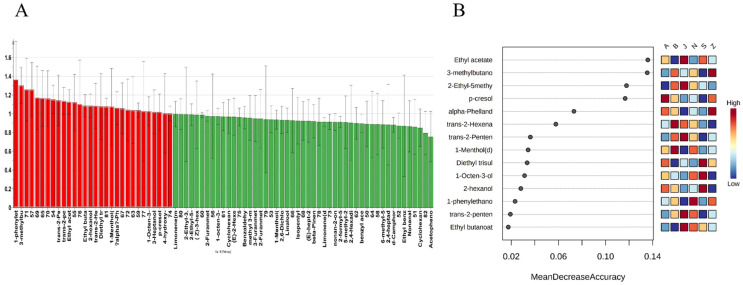
(**A**) Variable Importance for the Projection (VIP) diagram of the OPLS-DA model; (**B**) significant features identified by random forest.

**Table 1 molecules-27-04393-t001:** Identification of the VOCs in SS by HS-GC-IMS.

Compounds	Formula	RI ^1^	RT ^2^ [s]	DT ^3^ [ms]	CAS
Terpenoids					
limonene (d)	C_10_H_16_	1027.1	613.85	1.28735	138-86-3
limonene (m)	C_10_H_16_	1025.1	608.277	1.64831	138-86-3
d-camphor	C_10_H_16_O	1137.1	1000.58	1.84257	464-49-3
beta-pinene	C_10_H_16_	972.2	483.778	1.21748	127-91-3
alpha-phellandrene	C_10_H_16_	991.9	525.443	1.2135	99-83-2
1-menthol (m)	C_10_H_20_O	1172.8	1172.199	1.87632	2216-51-5
1-menthol (d)	C_10_H_20_O	1174.1	1178.76	1.25862	2216-51-5
Alcohols					
linalool	C_10_H_18_O	1102.8	858.912	1.21718	78-70-6
1-phenylethanol	C_6_H_8_O_3_	1059.5	708.841	1.19687	98-86-2
3-heptanol	C_7_H_16_O	901.5	359.303	1.3301	589-82-2
3-furanmethanol	C_5_H_6_O_2_	975.6	490.8	1.10433	4412-91-3
2-hexanol	C_6_H_14_O	813.5	254.742	1.57558	626-93-7
2-furanmethanol,5-methyl-(m)	C_6_H_8_O_2_	954.4	448.858	1.55977	3857-25-8
2-furanmethanol,5-methyl-(d)	C_6_H_8_O_2_	959.4	458.318	1.25549	3857-25-8
1-octen-3-ol	C_8_H_16_O	983.9	508.084	1.15751	3391-86-4
(Z)-3-hexen-1-ol	C_6_H_12_O	851.3	294.963	1.51428	928-96-1
Phenols					
p-cresol	C_7_H_8_O	1084.6	792.197	1.1635	106-44-5
2,6-dichlorophenol	C_6_H_4_Cl_2_O	1204.1	1347.154	1.20289	87-65-0
Aldehydes					
trans-2-Hexenal	C_6_H_10_O	825	266.352	1.52158	6728-26-3
trans-2-pentenal (m)	C_5_H_8_O	753.8	199.608	1.35623	1576-87-0
trans-2-pentenal (d)	C_5_H_8_O	753.8	199.608	1.10847	1576-87-0
Nonanal	C_9_H_18_O	1106.1	871.718	1.47192	124-19-6
5-methyl-2-furancarboxaldehyde	C_6_H_6_O_2_	936.7	416.718	1.1367	620-02-0
2-formyl-5-methylthiophene	C_6_H_6_OS	1117.4	916.415	1.15464	13679-70-4
2,4-heptadienal	C_7_H_10_N_2_	1001.5	547.827	1.62004	5910-85-0
(E)-hept-2-enal	C_7_H_12_O	959.3	458.146	1.66579	18829-55-5
benzaldehyde	C_7_H_6_O	959	457.688	1.14877	100-52-7
(E)-2-Hexenal	C_6_H_10_O	851.5	295.153	1.18069	6728-26-3
2,4-Hexadienal, (E,E)-	C_6_H_8_O	913.5	377.923	1.44546	142-83-6
Esters					
methyl 3-methylbutanoate	C_6_H_12_O_2_	764.7	209.149	1.5252	556-24-1
Isopentyl isovalerate	C_10_H_20_O_2_	1116.7	913.692	2.03768	659-70-1
ethyl butanoate	C_6_H_12_O_2_	790.2	232.779	1.55819	105-54-4
ethyl benzoate	C_9_H_10_O_2_	1142.7	1025.57	1.25881	93-89-0
ethyl acetate	C_4_H_8_O_2_	617.2	122.892	1.33577	141-78-6
benzyl acetate	C_9_H_10_O_2_	1128.1	961.307	1.32321	140-11-4
Acids					
3-methylbutanoic acid	C_5_H_10_O_2_	849	292.352	1.48126	503-74-2
Ketones					
nonan-2-one	C_9_H_18_O	1095.5	831.727	1.40358	821-55-6
cyclohexanone (m)	C_6_H_10_O	897.3	353.068	1.45503	108-94-1
cyclohexanone (d)	C_6_H_10_O	897.5	353.34	1.15396	108-94-1
Acetophenone	C_8_H_8_O	1048.6	675.279	1.17218	98-86-2
6-methyl-5-hepten-2-one	C_8_H_14_O	942.5	426.954	1.1742	110-93-0
Furaneol	C_6_H_8_O_3_	1015.3	582.545	1.61471	3658-77-3
Others					
2-Ethyl-5-methylpyrazine	C_7_H_10_N_2_	1006.3	559.704	1.66669	13360-64-0
2-Ethyl-3,5-dimethylpyrazine	C_8_H_12_N_2_	1076.4	764.142	1.73198	13925-07-0
diethyl trisulfide	C_4_H_10_S_3_	1126.2	952.929	1.24243	3600-24-6

^1^ Represents the retention index calculated using n-ketones C4-C9 as an external standard on the FS-SE-54-CB-1 column; ^2^ Represents the retention time in the capillary GC column; ^3^ Represents the drift time in the drift tube: (m): monomer; (d): dimer.

**Table 2 molecules-27-04393-t002:** The relative contents of VOCs in SS from different regions were determined by HS-SPME-GC-MS.

Code	Compounds	RT	CAS	Relative Content (%)					
		(Min)		B	N	A	Z	J	S
	Terpenoids								
1	myrcene	6.922	123-35-3	0.06 ± 0.01	0.05 ± 0.02	0.03 ± 0.02	0.04 ± 0.01	0.05 ± 0	0.02 ± 0.01
2	(−)-limonene	7.808	5989-54-8	0.46 ± 0.04	0.40 ± 0.07	0.19 ± 0.07	0.26 ± 0.14	0.38 ± 0.14	0.21 ± 0.05
3	l-menthone	11.003	14073-97-3	4.28 ± 0.51	4.83 ± 0.77	5.60 ± 0.96	4.92 ± 0.54	5.97 ± 0.47	5.80 ± 0.67
4	1,4-cyclohexadiene, 3-ethenyl-1,2-dimethyl-	7.257	62338-57-2	0.23 ± 0.03	0.22 ± 0.03	0.19 ± 0.04	0.16 ± 0.02	0.16 ± 0	0.19 ± 0.02
5	1,3,8-p-menthatriene	7.721	18368-95-1	0.28 ± 0.05	0.28 ± 0.04	0.18 ± 0.03	0.16 ± 0.02	0.16 ± 0.02	0.18 ± 0.01
6	(+)-pulegone	13.565	89-82-7	36.83 ± 3.16	33.76 ± 1.26	51.99 ± 5.54	51.00 ± 3.05	43.69 ± 0.93	44.78 ± 2.81
7	piperitone	13.797	89-81-6	2.46 ± 1.27	2.37 ± 0.52	0.84 ± 0.31	1.76 ± 0.29	1.83 ± 0.27	1.96 ± 0.22
8	*β*-bourbonene	18.752	5208-59-3	0.22 ± 0.02	0.21 ± 0.03	0.37 ± 0.07	0.21 ± 0.05	0.20 ± 0.02	0.32 ± 0.04
9	α-copaene	18.404	3856-25-5	0.22 ± 0.02	0.18 ± 0.02	0.36 ± 0.06	0.28 ± 0.08	0.19 ± 0.02	0.26 ± 0.04
10	cubebene	18.968	13744-15-5	ND	ND	0.09 ± 0.04	0.04 ± 0.03	ND	ND
11	caryophyllene	20.004	87-44-5	0.39 ± 0.09	0.34 ± 0.04	2.21 ± 1.04	0.96 ± 0.52	0.36 ± 0.03	0.72 ± 0.12
12	*β*-elemen	19.055	515-13-9	0.04 ± 0.01	ND	0.12 ± 0.07	0.03 ± 0.03	ND	0.03 ± 0.02
13	(±)-beta-copaene	22.044	18252-44-3	0.05 ± 0.03	0.04 ± 0.01	1.59 ± 1.05	0.27 ± 0.20	0.04 ± 0.02	0.15 ± 0.07
14	(−)-humulene epoxide II	25.868	19888-34-7	0.08 ± 0.02	0.10 ± 0	0.07 ± 0.03	0.05 ± 0.02	0.07 ± 0.02	0.09 ± 0.03
15	(+)-delta-Cadinene	23.359	483-76-1	0.06 ± 0.01	0.06 ± 0.01	0.23 ± 0.11	0.11 ± 0.08	0.04 ± 0	0.09 ± 0.01
16	menthofuran	11.269	494-90-6	1.02 ± 0.19	ND	ND	0.92 ± 0.13	ND	ND
17	(+)-isomenthone	14.213	1196-31-2	2.98 ± 0.13	3.15 ± 0.34	1.08 ± 0.57	2.00 ± 0.98	2.71 ± 0.21	1.91 ± 0.20
18	verbenone	15.194	18309-32-5	10.96 ± 1.70	10.85 ± 0.94	5.96 ± 1.42	8.00 ± 1.63	9.63 ± 0.41	8.79 ± 0.85
19	caryophyllene oxide	25.089	1139-30-6	1.32 ± 0.18	1.41 ± 0.08	0.94 ± 0.41	0.76 ± 0.10	1.10 ± 0.23	1.44 ± 0.41
	Alcohols								
20	(E)-p-mentha-2,8-dien-1-ol	10.52	7212-40-0	0.18 ± 0.06	0.16 ± 0.03	0.08 ± 0.02	0.10 ± 0.04	0.14 ± 0.01	0.15 ± 0.02
21	isopulegol	11.999	89-79-2	0.25 ± 0.04	0.29 ± 0.03	ND	0.17 ± 0.05	0.21 ± 0.02	0.13 ± 0.03
22	spathulenol	24.949	6750-60-3	0.13 ± 0.02	0.16 ± 0	0.13 ± 0.06	0.09 ± 0.02	0.11 ± 0.02	0.14 ± 0.07
23	2-cyclohexen-1-ol	12.294	74410-00-7	0.19 ± 0.10	0.21 ± 0.04	0.08 ± 0.05	0.10 ± 0.05	0.20 ± 0.03	0.22 ± 0.04
	Phenols								
24	Thymol	14.672	89-83-8	0.06 ± 0.01	0.08 ± 0.01	0.05 ± 0.02	0.06 ± 0.02	0.06 ± 0.01	0.04 ± 0.03
	Esters								
25	trans-carveyl acetate	16.712	1134-95-8	6.21 ± 0.53	6.12 ± 0.65	6.18 ± 0.40	5.53 ± 0.79	5.41 ± 0.07	6.01 ± 0.54
26	carveylacetate	16.949	97-42-7	0.28 ± 0.08	0.27 ± 0.05	0.24 ± 0.03	0.21 ± 0.02	0.19 ± 0	0.21 ± 0.04
27	nepetalactone	18.124	21651-62-7	0.61 ± 0.63	0.25 ± 0.06	0.08 ± 0.06	0.12 ± 0.03	0.14 ± 0.02	0.12 ± 0.02
28	dibutyl phthalate	31.553	84-74-2	0.15 ± 0.17	0.06 ± 0.01	0.04 ± 0	0.06 ± 0.01	0.09 ± 0.03	0.08 ± 0.01
	Ketones								
29	3-methyl-Cyclohexanone	6.029	591-24-2	0.27 ± 0.17	0.23 ± 0.08	0.13 ± 0.07	0.15 ± 0.08	0.20 ± 0.02	0.24 ± 0.03
30	2-isopropyl-2,5-dimethylcyclohexanone	12.758	20144-44-9	4.12 ± 0.41	4.06 ± 0.29	ND	2.85 ± 0.31	3.39 ± 0.14	3.23 ± 0.17
31	2-cyclopenten-1-one, 3-ethyl-2-hydroxy-	8.335	21835-01-8	0.30 ± 0.03	0.27 ± 0.01	0.07 ± 0.06	0.20 ± 0.07	0.24 ± 0.09	0.14 ± 0.03
32	trans-isopulegone	11.588	29606-79-9	1.02 ± 0.08	0.93 ± 0.05	1.12 ± 0.09	0.97 ± 0.07	1.13 ± 0.03	1.05 ± 0.08
33	berbenone	12.534	80-57-9	0.46 ± 0.06	0.52 ± 0.02	0.29 ± 0.06	0.29 ± 0.16	0.50 ± 0.03	0.40 ± 0.04
34	piperitenone	17.128	491-09-8	3.08 ± 0.14	3.40 ± 0.07	3.31 ± 0.07	3.35 ± 0.15	2.81 ± 0.15	3.16 ± 0.35
35	jasmone	19.347	488-10-8	0.10 ± 0.02	0.10 ± 0.01	0.11 ± 0.02	0.08 ± 0.02	0.07 ± 0.01	0.08 ± 0.01
36	2-isopropylidene-5-methylcyclohexanone	20.63	15932-80-6	1.27 ± 0.77	2.34 ± 1.34	0.45 ± 0.44	0.54 ± 0.21	0.81 ± 0.26	0.29 ± 0.12
37	cyclohexanone, 2-(2-butynyl)-	14.459	54166-48-2	0.27 ± 0.02	0.29 ± 0.03	0.23 ± 0.02	0.26 ± 0.01	0.22 ± 0.01	0.17 ± 0.07
38	cinerolone	19.429	17190-74-8	0.14 ± 0	0.18 ± 0.02	0.11 ± 0.04	0.12 ± 0.02	0.09 ± 0	0.17 ± 0.01
39	trans-Pulegone oxide	13.125	13080-28-9	0.38 ± 0.10	0.39 ± 0.13	ND	ND	0.32 ± 0.04	ND
	Others								
40	1-methyl-3-prop-1-en-2-ylbenzene	9.345	1124-20-5	0.17 ± 0.02	0.15 ± 0.01	0.15 ± 0.05	0.16 ± 0.02	0.15 ± 0.02	0.07 ± 0.05
41	dehydroxymenthofurolactone	22.721	38049-04-6	4.51 ± 1.41	5.09 ± 0.90	2.22 ± 1.21	2.21 ± 1.05	3.66 ± 0.53	4.08 ± 0.70
42	isomintlactone	23.586	75684-66-1	0.83 ± 0.22	0.95 ± 0.10	0.39 ± 0.23	0.40 ± 0.17	0.69 ± 0.08	0.67 ± 0.08

“ND” means undetected or content less than 0.01%.

## Data Availability

The data presented in this study are available in Appendix A.

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
