# Peer review of "Discrimination and Characterization of the Volatile Organic Compounds in Schizonepetae Spica from Six Regions of China Using HS-GC-IMS and HS-SPME-GC-MS"

_molecules, 2022, doi:10.3390/molecules27144393_

Round 1

Reviewer 1 Report

The paper intend to explore the VOCs of Schizonepetae Spica from 6 regions of China using two different technologies (HS-SPME/GC-MS and HS/GC/IMS) and the authors conclude on the superiority of HS/GC/IMS for identification of TCM of different origin.

-However to be able to compare efficiently the two technologies, a deeper study on HS-SPME/GC-MS conditions should have been performed : In paragraph 4.4 it is only indicated : "SPME fiber (...) as quality control for optimization by comparing the results of GC-MS under different SPME conditions. The final conditions are as follows: Samples in headspace bottles were at pre-equilibrated 60°C and extracted with a PDMS/CAR/DVB fiber column at the same temperature for 25 min."

Different fibers should have been tested in order to select the one that will allow for the maximum extraction of volatile compounds. Temperature and time parameters for extraction should also have been taken into account. A method should have been developped varying the parameters.

-In the introduction, literature data, as complete and precise as possible (artcles, reviews, books, etc.), about VOCs already known in SS should be given (families of compounds, methods for extraction and identification, if any).

-Abstract and conclusion : 82 VOCs have been detected (but not identified as written in the abstract : this should be changed), however, only 40 compounds where identified by HS/GC/IMS (see paragraph 2.1.1)(line 96) . Moreover 45 compounds are present in Table 1 (identification by HS-GC-IMS). And 42 compounds where identified by HS-SPME/GC-MS (paragraph 2.2)(line 134) and presumably Table 2 (this should be indicated in the legend of table 2).

-In table 1 (column 1 : "compounds") what do "(d)" and "(m)" mean for some compounds ?

-In table 1 (column 1 : "compounds") : "Aldehydes", "Ketones" should be in bold letters

-In table 1 retention time (column 4) : what is the unit ? It should be given in minutes (decimal figure and decimal point).

In Materials and Methods  (chemical and reagents) it seems that only 6 analytical standards were used ? and CAS 89-81-6 is piperitone not (+)-pulegone. All  commercially available analytical standards should be used to make and confirm the identifications.

Therefore the studies performed here are too partial to conclude on the superiority of one method over another and more generally on the use of HS/GC/IMS of TCM from different origins.

-line 31 : "Schizonepeta tenuifolia Briq" : Briq sould not be in italic

-line 32 : general refs (reviews, books, monography...) about SS in TCM sould be given

-line 36 : "The most important active ingredients are volatile organic compounds" (plural form)

-line 37-39 : "According to existing reports,...". What are these reports ? (refs needed).

-line 51-53 : "With the development  of.... traditional Chinese medicine" : references are needed.

-line 61 : "At the same time, SPME has been proved to be a simple, rapid, sensitive and solvent-free method suitable for determination of VOCs in TCMs [8]". But ref [8] does not concern TCM plant but a plant from Algeria. A ref should correspond to an accurate information.

-line 63-64 : "HS-GC-IMS usually detects small molecules of volatile components, which makes up for the deficiency of HS-SPME-GC-MS that can only capture medium molecules of volatile components [9]". In ref 9 it is not indicated such a general conclusion. Moreover it depends on the fiber, temperature and time of extraction for HS-SPME.

-lines 75-76 : "The difference between this study and other studies published in the literature is that two different hardware methods are compared using the same data processing program.". Refs (and details of methods) about these studies should be given.

-Figure 2 is not easy to read.

-Line 113-114 : "are also differences among different batches within the same region, which may contribute to their different storage methods.". What are these methods ? reference ?

-Table 2 : Relative content (%). Percentage of what ?

-All Figures : too small and blurry. Some figures or words are un-readable.

-line 325 : CAS 89-81-6 is piperitone and not (+)-pulegone. Please check all CAS numbers.

-line 326 : what is "caryophyllene oxidewere" ? blanck between "oxide" and "were" ? The verb "were" is probably at the wrong place.

-lines 326 : "and pulegonev (CAS:89-82-7, ≥98%)" CAS 89-82-7 is pulegone (not "pulegonev")

-In conclusion there is nothing about the numbers of compounds identified using both technology.

Author Response

Response to Reviewer 1 Comments

Point 1: Different fibers should have been tested in order to select the one that will allow for the maximum extraction of volatile compounds. Temperature and time parameters for extraction should also have been taken into account. A method should have been developped varying the parameters.

Response 1: We thank the reviewer and agree on this point. We have added the method of experimental condition optimization. SPME fiber (supelco, Bellefonte, Penn.) was installed on a MultiPurpose sampler (Ger-stel, GER) and combined with 7890B-7000D triple quadrupole gas chromatography mass spectrometry (Agilent Technologies, Palo Alto, CA, USA) to detect VOCs in SS samples. The univariate method was used to select the SPME conditions for each factor indi-vidually. Crushed samples (0.1 grams) were placed into a 20 mL headspace vial with a magnetic screw cap and a Teflon-lined rubber septum. Then, the sample vial was equilibrated for 5 minutes at a certain temperature (50 °C, 60 °C, 70 °C, and 80 °C) on a heating platform. The extraction was conducted by inserting the SPME fibers (poly-acrylate 85 µm, polydimethylsiloxane/divinylbenzene 65 µm phase thickness (PDMS/DVB), polydimethylsiloxane/carbon wide range/divinylbenzene 50/30 µm phase thickness (PDMS/CAR/DVB)) into the head space of the vial for a certain time (15min, 20 min, 25 min and 30min). The fiber was desorbed into the injection port of the GC for 5 minutes at the end of the extraction. Under the same conditions of sample size, extrac-tion temperature and extraction time, the optimal SPME fiber was first determined. Then the best SPME fiber was chosen to optimize the other factors at the same way, and peak capacity was used as the criterion. The resulted optimal extraction parameters were determined as follows, using PDMS/CAR/DVB fiber to extract 0.1 g of sample at 60 °C for 25 min.

Point 2: In the introduction, literature data, as complete and precise as possible (artcles, reviews, books, etc.), about VOCs already known in SS should be given (families of compounds, methods for extraction and identification, if any).

Response 2: We thank the reviewer for the query. In the revised version, we have supplemented the description and literature support of this part in the introduction. Chemical studies showed that SS contained volatile organic compounds (VOCs), fla-vonoids and organic acids, among which VOCs was the main medicinal component, and pharmacological activities of VOCs possesses anti-inflammatory, antineoplastic and antiviral properties (PloS one, 2020, 15(1), e0227235. Nat Prod Res Dev, 2020, 32: 1087-1098. J Ethnopharmacol. 2016, 194:580-586. Acta Pharm Sin. 2017, 52(1):126-131. China J Chin Mater Med. 2017; 42(9):1717-1721. J Microbiol. 2018, 56(9): 683-689). Studies on SS mainly focus on volatile oil extraction and bioactive ingredients, but the overall characterization of SS from different sources is not comprehensive enough( Biomedical Chromatography 2021, 35 , 5106.).

Point 3: Abstract and conclusion : 82 VOCs have been detected (but not identified as written in the abstract : this should be changed), however, only 40 compounds where identified by HS/GC/IMS (see paragraph 2.1.1)(line 96) . Moreover 45 compounds are present in Table 1 (identification by HS-GC-IMS). And 42 compounds where identified by HS-SPME/GC-MS (paragraph 2.2)(line 134) and presumably Table 2 (this should be indicated in the legend of table 2).

Response 3: We thank the reviewer for the query. In the revised version, we revised the abstract and re-calibrated the legend of table 2. The number of compounds detected by HS-GC-IMS is explained as follows: In HS-GC-IMS, since monomer ions and neutral molecules might form adjunct substances in the drift region, several single compounds might produce multiple signals so that the same substance could detect monomers or dimers. In Table 1, (m) and (d) represent monomers and dimers respectively. Therefore, 40 compounds are present in Table 1.

Point 4: In table 1 (column 1 : "compounds") what do "(d)" and "(m)" mean for some compounds ?

Response 4: Thank you for your suggestion. In Table 1, (m) and (d) represent monomers and dimers respectively.

Point 5:In table 1 (column 1 : "compounds") : "Aldehydes", "Ketones" should be in bold letters.

Response 5: Thank you for your suggestion. In table 1, we have indicated  "Aldehydes", "Ketones" in bold letters.

Point 6:In table 1 retention time (column 4) : what is the unit ? It should be given in minutes (decimal figure and decimal point).

Response 6: Thank you for your suggestion. The unit of the retention time of HS-GC-IMS is second.

Point 7: In Materials and Methods  (chemical and reagents) it seems that only 6 analytical standards were used ? and CAS 89-81-6 is piperitone not (+)-pulegone. All commercially available analytical standards should be used to make and confirm the identifications.

Response 7: We appreciate the reviewer and agree on this point. In the revision process, the name of analytical standards was carefully checked and corrected. The modification result is as follows:piperitone (CAS:89-81-6) and (+)-pulegone (CAS:89-82-7). We agree with the recommendation that all commercially available analytical standards should be used to make and confirm the identification. And we'll pay attention to that in future experiments. But we have carefully assessed the funding and experimental conditions required to complete these additional studies and feel that we cannot afford to extend this scope at present. In this study, the six analytical standards were chosen for the following reasons: (+)-pulegone, piperitone, and caryophyleneoxide were the main components contributing to the antioxidant and anti-inflammatory activities in SS (Biomed Chromatogr. 2021;35(7):e5106.). Caryophyllene oxidewere also has anti-inflammatory activity. Limonene has antibacterial properties. We identified the other compounds by combining three methods, including searching for spectral peaks in the NIST17 standard mass spec-trometry library of the chemistry workstation, calculating retention index (RI) values, and comparing the retention times of substances in the sample with external standards.

Point 8: line 31 : "Schizonepeta tenuifolia Briq" : Briq sould not be in italic.

Response 8: Thanks to this reviewer for the careful checking work. We have corrected in the revised version.

Point 9: line 32 : general refs (reviews, books, monography...) about SS in TCM sould be given.

Response9: We thank the reviewer for the query. In the revised version, we have supplemented the description and literature support of this part in the introduction.

Point 10:line 36 : "The most important active ingredients are volatile organic compounds" (plural form)

Response10: We appreciate the reviewer for this suggestion. We have corrected in the revised version.

Point 11:-line 37-39 : "According to existing reports,...". What are these reports ? (refs needed).

Response11: Thanks to this reviewer for the careful checking work. We have corrected in the revised version. Chemical studies showed that SS contained volatile organic compounds (VOCs), flavonoids and organic acids, among which VOCs was the main medicinal component(PloS one, 15(1), e0227235.).

Point 12: line 51-53 : "With the development  of.... traditional Chinese medicine" : references are needed.

Response12:Thanks to this reviewer for the careful checking work. In the revised version, we have supplemented the literature support of this part.

Point 13: line 61 : "At the same time, SPME has been proved to be a simple, rapid, sensitive and solvent-free method suitable for determination of VOCs in TCMs [8]". But ref [8] does not concern TCM plant but a plant from Algeria. A ref should correspond to an accurate information.

Response13: We appreciate the reviewer′s comment on the references. In the revised version, we have revised this sentence and modified the reference. And we carefully checked other references to ensure that a reference correspond to an accurate information.

Point 14: line 63-64 : "HS-GC-IMS usually detects small molecules of volatile components, which makes up for the deficiency of HS-SPME-GC-MS that can only capture medium molecules of volatile components [9]". In ref 9 it is not indicated such a general conclusion. Moreover it depends on the fiber, temperature and time of extraction for HS-SPME.

Response14: We thank the reviewer for the query. According to your suggestion, we have modified this sentence and the reference to make the article more rigorous.

Point 15: lines 75-76 : "The difference between this study and other studies published in the literature is that two different hardware methods are compared using the same data processing program.". Refs (and details of methods) about these studies should be given.

Response15: We thank the reviewer for the query. We have modified this sentence. And reflected in page2, line 77-79.

Point 16: Figure 2 is not easy to read.

Response16: We feel sorry for the unclear expressions of Figure 2. We adjusted Figure 2 to make it easier to read. The top view of GC-IMS 3D topographic plots of SS samples were shown in Figure2A. The entire spectrum represents the total number of compounds extracted from the samples. Each point on the right side of RIP represents a volatile compound in the samples, and the color of the point represents the concentration of the substance. The darker the color, the higher the density. In order to further compare the differences of volatile compounds in SS of different regions, the sample information was presented in the form of fingerprint (Figure2-B). The fingerprint contains all signals that can be detected in the instrument, where each row represents a sample and each column represents a substance. Known substances were marked with existing names and unknown substances were marked with numbers(Figure2C).

Point 17:Line 113-114 : "are also differences among different batches within the same region, which may contribute to their different storage methods.". What are these methods ? reference ?

Response17: Thank you for your suggestion. According to your suggestion, we have modified this sentence and supplemented the literature support of this part. And reflected in page5, line 115-118.

Point 18: Table 2 : Relative content (%). Percentage of what ?

Response18: We feel sorry for the unclear expressions. In this experiment, for data processing of HS-SPME-GC-MS results, we refer to the study of Chen et al. Chen et al. studied finger citron (Citrus medica L.var. sarcodactylis) based on HS-SPME-GC-MS, used the peak area normalization method to perform relative quantitative calculation to obtain the relative percentage content of each compound in finger citron samples of different pickling steps. The percentage represents the ratio of individual compounds to the total peak area of the sample.

Point 19 : All Figures : too small and blurry. Some figures or words are un-readable.

Response19: Thanks to the reviewer for suggestion. We have adjusted the table in the article.

Point 20 : line 325 : CAS 89-81-6 is piperitone and not (+)-pulegone. Please check all CAS numbers.

Response20: Thanks to the reviewer for the careful checking work. In the revision process, the name of analytical standards was carefully checked and corrected. The modification result is as follows:piperitone (CAS:89-81-6) and (+)-pulegone (CAS:89-82-7).

Point 22 : line 326 : what is "caryophyllene oxidewere" ? blanck between "oxide" and "were" ? The verb "were" is probably at the wrong place.

Response22: We appreciate the reviewer′s comment on the grammar. In the revision process, the name of analytical standards was carefully checked and corrected.

Point 23 : lines 326 : "and pulegonev (CAS:89-82-7, ≥98%)" CAS 89-82-7 is pulegone (not "pulegonev")

Response23: Thanks to this reviewer for the careful checking work. In the revision process, the name of analytical standards was carefully checked and corrected.

Reviewer 2 Report

Comments

The proposed methodology is not a novel one, however the combinatory use of the two techniques (i.e. HS-GC-IMS and HS-SPME-GC-MS) for application in this particular type of chinese medicinal herbs has a reasonable innovation.

The authors provided a thorough analysis of VOCs and achieved a successful statistical discrimination of the origin of schizonepetae spicae from 6 regions in China. The Conclusions are supported by the analytical results. Unfortunately, authors did not included in the research a discrimination of VOCs for the herb and the spica of schizonepetae as far as I understand, which is also of great consumer's interest since the difference in the medicinal value of these two parts remains controversial nowadays and it could add more value to this paper.

Overall, if some minor issues listed below are resolved, I would recommend acceptance of the paper.

Line 95. It is written "retention times, drift times, and Ko" for Table 1. However, what is the meaning of Ko which does not appear at all in Table 1?

Although some information is given in the text, please provide explanation of low-case “a” and “b” appearing in the charts, except of the upper case A,B and C.

In Table 2, the numbers of relative content are confused and must be clearly separated.

Line 187. It is written “Then, based on analytical methods of both HS-GC-IMS and HS-SPME-GC-MS, 5 predictive components and 8 orthogonal components were selected for the OPLS-DA model.” Give more explanation about the criteria of the selection of these 5 and 8 components.

Fig. 4. Legend: please check if A,B,C,D are described correctly. I suggest to use the term “Score plots” for the left charts and the “permutation test” for the right ones with correcting the referring letters. Additionally, I have to say that the achieved clustering illustrated in score plots is impressively successful, although usually there are some aliases and experimental deviations.

Reviewer 3 Report

According to the authors, “The difference between this study and other studies published in the literature is that two different hardware methods are compared using the same data processing program”. These “other studies” are not referenced, though, hence it is hard to tell how original the contribution really is.

I very much doubt that VOCs are the main medicinal components of TCM. Of course when one uses methods that can only detect VOCs, they may seem responsible for the medicinal properties, but I am quite confident that there are many much more biologically active compounds that can only be determined using liquid phase separations. Please address this.

The number of samples from any given region was very small (3). Any claims of regional differences based on such a small number of samples are very dubious, as these differences might be completely accidental. The authors tacitly admit as much in the last sentence of the conclusions. In my opinion, rather than claiming regional differences, the authors should just talk about the differences between the samples.

Not all experimental details are clear. It is not explained why flow programming was used with the HS-GC-IMS system. The samples were “incubated at 75 °C for 20 min at 500 r/min” – what does 500 r/min refer to?  It is unclear if other SPME fibers were evaluated, and if yes, which ones? Was the triple quad instrument used as a single quad in the experiments?

SPME extraction was carried out at 60 °C. While this improves the kinetics of the extraction process, it reduces the sensitivity of the method because partition coefficients decrease when the temperature increases. This is particularly important for more volatile compounds with lower partition coefficients, which explains why the authors could not detect them with GC-MS. This should be explained in the text.

Peak area normalization method is a poor choice for the determination of the relative content of each compound with PDMS/CAR/DVB SPME fibers. These fibers extract analytes through adsorption, which is a competitive process. A given peak area in this case depends not only on the concentration of a particular analyte, but also on the extracted amount of all other compounds, hence it might differ from sample to sample even if the relative content is in fact the same. This must be discussed.

In lines 378-9, the authors wrote: “The HS-SPME-GC-MS results were expressed as the mean ± standard deviation” – the mean of what?

The quality of English must be improved. Already the first sentence of the Introduction contains an error (should be “dry spike”, not “dry Spica”). Apart from obvious errors due to lack of knowledge of the correct terminology (e.g. “medium molecules”, ”exterior syndrome”, “compounds driven into an ionization chamber”, “headspace bottles”, “PDMS/CAR/DVB fiber column”, “under the 230°C hot stripping”, “elastic quartz capillary column”, “heating procedure of the column”, etc.), there are numerous grammatical errors that must be fixed.

On p. 11, the authors wrote: “By using the well-established HS-GC-IMS technique, 40 VOCs were discovered in SS. On the other hand, 42 VOCs were identified in SS, by the established HS-GC-IMS analysis method.” This is obviously wrong.

Section 2.1.1 starts with the sentence: “The VOCs of the SS in six different regions were also analyzed by HS-GC-IMS”. Why “also” if this is the first method discussed?

Section 2.4 starts with the sentence: “(…) due to the influence of growing environment, the composition and content of volatile components in different citrus teas are also different”. How is this relevant to the paper? No citrus teas were analyzed in it.

The sentence in line 352-3 should be deleted, as it is just an incorrect version of the next sentence.

Author Response

Response to Reviewer 3 Comments

Point 1: According to the authors, “The difference between this study and other studies published in the literature is that two different hardware methods are compared using the same data processing program”. These “other studies” are not referenced, though, hence it is hard to tell how original the contribution really is.

Response 1: We thank the reviewer for the query. This sentence is not precise enough,so we have changed it in the revision process. And reflected in page2, line 101-105.

Point 2: I very much doubt that VOCs are the main medicinal components of TCM. Of course when one uses methods that can only detect VOCs, they may seem responsible for the medicinal properties, but I am quite confident that there are many much more biologically active compounds that can only be determined using liquid phase separations. Please address this.

Response 2: We appreciate the reviewer for this suggestion. There are indeed many biologically active compounds in TCM that can only be identified by liquid phase separations. Chemical studies showed that Schizonepetae Spica contained volatile organic compounds (VOCs), flavonoids and organic acids, among which VOCs was the main medicinal component, and pharmacological activities of VOCs possesses anti-inflammatory, antineoplastic and antiviral properties (PloS one, 2020, 15(1), e0227235. Nat Prod Res Dev, 2020, 32: 1087-1098. J Ethnopharmacol. 2016, 194:580-586. Acta Pharm Sin. 2017, 52(1):126-131. China J Chin Mater Med. 2017; 42(9):1717-1721. J Microbiol. 2018, 56(9): 683-689). However, we accept your suggestion and will pay attention to other chemical components of TCM in future studies.we have changed the first sentence of the abstract. And reflected in page1, line 18-19.

Point 3: The number of samples from any given region was very small (3). Any claims of regional differences based on such a small number of samples are very dubious, as these differences might be completely accidental. The authors tacitly admit as much in the last sentence of the conclusions. In my opinion, rather than claiming regional differences, the authors should just talk about the differences between the samples.

Response 3: Thank you for this valuable feedback. Xiao et al. also purchased three batches from one region for each sample in the study of characterization of key odor components of fennel essential oil in different regions (J Chromatogr B Analyt Technol Biomed Life Sci. 2017;1063:226-234.). Therefore, in this study, three batches of medicinal materials were purchased from each region. However, we accept your suggestion and will pay attention to it in future experiments. In this article, we have made some changes according to your suggestions. And reflected in page1, line25-26; page2, line101-105 and page14, line611-612.

Point 4: Not all experimental details are clear. It is not explained why flow programming was used with the HS-GC-IMS system. The samples were “incubated at 75 °C for 20 min at 500 r/min” – what does 500 r/min refer to?  It is unclear if other SPME fibers were evaluated, and if yes, which ones? Was the triple quad instrument used as a single quad in the experiments?

Response 4: We thank the reviewer and agree on this point. The HS-GC-IMS has an incubator, 500 r/min refers to the incubator speed.We have added the method of experimental condition optimization as the reviewer suggested. And reflected in page13, line 358-376. In this experiment, three kinds of fibers were tested: polyacrylate 85 µm, polydimethylsiloxane/divinylbenzene 65µm phase thickness (PDMS/DVB), polydimethylsiloxane/carbon wide range/divinylbenzene 50/30 µm phase thickness (PDMS/CAR/DVB). In the experiment, ions are scanned through the first quadrupole and then through the second quadrupole (no voltage applied)

Point5: SPME extraction was carried out at 60 °C. While this improves the kinetics of the extraction process, it reduces the sensitivity of the method because partition coefficients decrease when the temperature increases. This is particularly important for more volatile compounds with lower partition coefficients, which explains why the authors could not detect them with GC-MS. This should be explained in the text.

Response 5: We thank the reviewer for the query. In the revised version, we have supplemented the description in the discussion. And reflected in page11, line 427-429.

Point 6: Peak area normalization method is a poor choice for the determination of the relative content of each compound with PDMS/CAR/DVB SPME fibers. These fibers extract analytes through adsorption, which is a competitive process. A given peak area in this case depends not only on the concentration of a particular analyte, but also on the extracted amount of all other compounds, hence it might differ from sample to sample even if the relative content is in fact the same. This must be discussed.

Response 6: We thank the reviewer for the valuable feedback. In this experiment, we refer to the study of Chen et al. They studied finger citron (Citrus medica L.var. sarcodactylis) based on HS-SPME-GC-MS, also used the peak area normalization method to perform relative quantitative calculation to obtain the relative percentage content of each compound in finger citron samples of different pickling steps.

Point 7: In lines 378-9, the authors wrote: “The HS-SPME-GC-MS results were expressed as the mean ± standard deviation” – the mean of what?

Response 7: We thank the reviewer for the query. In the experiment, each region had three batches of medicinal materials, and the average value was the average of the relative content of these three batches of medicinal materials.

Point 8: The quality of English must be improved. Already the first sentence of the Introduction contains an error (should be “dry spike”, not “dry Spica”). Apart from obvious errors due to lack of knowledge of the correct terminology (e.g. “medium molecules”, ”exterior syndrome”, “compounds driven into an ionization chamber”, “headspace bottles”, “PDMS/CAR/DVB fiber column”, “under the 230°C hot stripping”, “elastic quartz capillary column”, “heating procedure of the column”, etc.), there are numerous grammatical errors that must be fixed.

Response 8: We appreciate the reviewer’s comment on the grammar mistakes. In the revision process, the manuscript was carefully checked and all careless errors and grammatical mistakes have been corrected.

Point 9: On p. 11, the authors wrote: “By using the well-established HS-GC-IMS technique, 40 VOCs were discovered in SS. On the other hand, 42 VOCs were identified in SS, by the established HS-GC-IMS analysis method.” This is obviously wrong.

Response9: We appreciate the reviewer’s comment on the grammar. We have corrected this mistake in the revision. And reflected in page12, line450.

Point 10: Section 2.1.1 starts with the sentence: “The VOCs of the SS in six different regions were also analyzed by HS-GC-IMS”. Why “also” if this is the first method discussed?

Response10: We appreciate the reviewer′s comment on the grammar. We have corrected this mistake in the revision.

Point 11: Section 2.4 starts with the sentence: “(…) due to the influence of growing environment, the composition and content of volatile components in different citrus teas are also different”. How is this relevant to the paper? No citrus teas were analyzed in it.

Response 11: We feel sorry for the unclear expressions. We have corrected this mistake in the revision. And reflected in page10, line360-361.

Point 12: The sentence in line 352-3 should be deleted, as it is just an incorrect version of the next sentence.

Response 12: We appreciate the reviewer′s comment on the language. We have corrected this mistake in the revision.

Reviewer 4 Report

The paper describes the important matter that Chinese medicines and schizonepetae spica, in particular need to be regulated, to ensure maximum efficacy for their use. Hence this paper assesses chemical methods to analyse  plant products with the aim  to find the better method and also to assess differences in plant concentration for different areas in China. I think there could be greater emphasis in the introduction that the paper's purpose is a quality control method for this particular Chinese herb, if that indeed is the case.

Line 37 states "according to existing reports... this needs references added.

Line 50 states GCxGC MS is not as widely used as GCMS...I suggest that the reason for this is added, such as much greater expense, increased complexity.

Line 324 space needed

Line 333 explain more about 500r/min

Line 323, here and elsewhere for limonene please be consistent with nomenclature, as regards chirality, m limonene? Check also for camphor etc

Table 1, Aldehydes should be in bold and also Ketones

Table 2, For % relative content, there are no spaces and its hard to read. I know the reason was probably because of an attempt to keep within the page margins, however I suggest the authors see whether the table could benefit by being in landscape format to give more room for spaces.

Line 274 spelling, it should be were

In the discussion, could the authors have a greater discussion on the reasons for differences in compounds and concentrations for key VOCs? This could include age of plants, sunlight differences, rainwater differences, soil differences transportation times? Were transportation times same and transportation conditions approx. similar? Please add transportation conditions.

In the discussion/introduction discuss whether there would be advantages/ disadvantages for using steam distillation.

Also discuss the literature on HPLC-MS, and what compounds could be found and how it compares with the work in this paper. The paper would benefit by mention of this, please see Chinese Pharmacopoeia, 2015 and also  discuss more findings in  Liu X. et al, Plos 1 , 2020

 In the method section, please add how the samples were powdered.

Is there any data available on how the samples were dried? Was any measurement undertaken on water content? Samples were said to be stored at 20oC, is this in the dark, please add if so?

 As regards the method, HS-GC-IMS, please discuss column temperature a bit. It seems a bit low?  Can you be sure that all the compounds are removed from the column? There isn’t a risk of carry over into the next sample? Was the low column temperature the reason why less high b.p. compounds were found compared to GCMS?

 Previous researchers have found 4,5,6,7-tetrahydro-3,6-dimethylbenzofuran. Do you think this could be present in your samples? Add a comment in the paper?

In summary, this is an interesting paper.

Author Response

Response to Reviewer 4 Comments

Point 1: Line 37 states "according to existing reports... this needs references added.

Response 1: We appreciate the reviewer′s comment on the references. In the revised version, we have supplemented the literature support of this part.

Point 2: Line 50 states GCxGC MS is not as widely used as GCMS...I suggest that the reason for this is added, such as much greater expense, increased complexity.

Response 2: We feel sorry for the unclear expressions. In the revised version, we added this part of content in introdruction. GC × GC is an analytical method with powerful separation function, which is often used by the sample analysis of complex ingredients. However, GC× GC is not as widely used as GC-MS due to its high cost and complex experimental operation.

Point 3: Line 324 space needed

Response 3: Thanks to this reviewer for the careful checking work. We have corrected this mistake in the revision.

Point 4: Line 333 explain more about 500r/min

Response 4: We thank the reviewer for this query. The HS-GC-IMS has an incubator, 500 r/min refers to the incubator speed.

Point 5: Line 323, here and elsewhere for limonene please be consistent with nomenclature, as regards chirality, m limonene? Check also for camphor etc.

Response 5: We would like to present sincere thanks to the reviewer for the reviewing work on our manuscript. In the revision process, the name of analytical standards was carefully checked and corrected. In HS-GC-IMS, since monomer ions and neutral molecules might form adjunct substances in the drift region, several single compounds might produce multiple signals so that the same substance could detect monomers or dimers.In Table 1, (m) and (d) represent monomers and dimers respectively.

Point 6: Table 1, Aldehydes should be in bold and also Ketones

Response 6: We appreciate the reviewer′s comment on the Table. In table 1, we have indicated  "Aldehydes", "Ketones" in bold letters.

Point 7: Table 2, For % relative content, there are no spaces and its hard to read. I know the reason was probably because of an attempt to keep within the page margins, however I suggest the authors see whether the table could benefit by being in landscape format to give more room for spaces.

Response 7: We appreciate the reviewer′s comment on the Table. In the revision, we have modified the margins in Table 2.

Point 8: Line 274 spelling, it should be were

Response 8: We appreciate the reviewer′s comment on the language. We have corrected this mistake in the revision.

Point 9: In the discussion, could the authors have a greater discussion on the reasons for differences in compounds and concentrations for key VOCs? This could include age of plants, sunlight differences, rainwater differences, soil differences transportation times? Were transportation times same and transportation conditions approx. similar? Please add transportation conditions.

Response 9: Thanks to this reviewer for the valuable feedback. In the revised version, we added this part of content in discussion. In addition, the two instruments reflect a certain difference in the content of volatile components of SS in different places, which may be related to factors such as climate conditions, soil conditions, sunshine intensity, cultivation conditions, and transportation conditions.

Point 10: In the discussion/introduction discuss whether there would be advantages/ disadvantages for using steam distillation.

Response 10: We appreciate the reviewer for this suggestion. In the revised version, we added this part of content in discussion. Steam distillation, solvent extraction and headspace capture are commonly used to extract volatile components from plants .However, the first two methods require large sample size, long extraction time and long heating time, and some components may be destroyed in the heating process. 

Point 11: Also discuss the literature on HPLC-MS, and what compounds could be found and how it compares with the work in this paper. The paper would benefit by mention of this, please see Chinese Pharmacopoeia, 2015 and also  discuss more findings in  Liu X. et al, Plos 1 , 2020.

Response 11: We appreciate the reviewer for this suggestion. There are indeed many biologically active compounds in TCM that can only be identified by liquid phase separations. Schizonepetae Spica contained volatile organic compounds (VOCs), flavonoids and organic acids, but VOCs was the main medicinal component. Therefore, this study focused on the volatile components of Schizonepetae Spica.

In the revision, we added this reference to the introduction.

Point 12: In the method section, please add how the samples were powdered.

Response 12: Thank you for this valuable feedback. In the revised version, we added this part of content in the method section. All SS samples were were crushed with a grinder (Tai site, Tianjin) and sieved through a 40-mesh sieve.

Point 13: Is there any data available on how the samples were dried? Was any measurement undertaken on water content? Samples were said to be stored at 20oC, is this in the dark, please add if so?

Response 13: We thank the reviewer for the query. All samples are dried Chinese medicinal materials purchased. Therefore, we did not measure the water content of the sample. However, we accept your suggestion and pay attention to this in future experiments. In the revision, we changed the storage conditions of the samples. The powdered sample was immediately packed in a plastic bag and stored in a dark, dry environment of 20 °C.

Point 14: As regards the method, HS-GC-IMS, please discuss column temperature a bit. It seems a bit low?  Can you be sure that all the compounds are removed from the column? There isn’t a risk of carry over into the next sample? Was the low column temperature the reason why less high b.p. compounds were found compared to GCMS?

Response 14: We thank the reviewer for the query. HS-GC-IMS with constant temperature chamber, generally do not adjust the column temperature. We reviewed three references(Poult Sci. 2020;99(12):7192-7201; Food Research International 2020, 138, 109717; Food Chem. 2022;375:131671.), all of which adopted a column temperature of 60 °C. In this experiment, two needle gaps were added between each sample to ensure that the compounds were removed from the column.

Point 15: Previous researchers have found 4,5,6,7-tetrahydro-3,6-dimethylbenzofuran. Do you think this could be present in your samples? Add a comment in the paper?

Response 15: Thank you for this valuable feedback. We looked up 4,5,6, 7-Tetrahydro-3,6-dimethylbenzofuran and the CAS number was 494-90-6. We detected this substance in HS-SPME-GC-MS, as shown in code 16 in Table 2.

Reviewer 5 Report

In this study, the authors compare the effectiveness of HS-SPME-GC-MS (HeadSpace Solid-Phase MicroExtraction-Gas Chromatography-Mass Spectrometry) and HS-GC-IMS (HeadSpace-Gas Chromatography-Ion Mobility Spectrometry) coupled to the same chemometric methods to classify Schizonepetae Spica from six different regions. By the two analytical techniques a total of 82 volatiles were identified. The regional classification using Orthogonal Partial Least Squares Discriminant Analysis (OPLS-DA) shows that the HS-GC-IMS method can classify samples better than the HS-SPME-GC-MS and has also the advantage of direct comparison of VOCs in different samples by data visualization.

Comments:

Line 22: please eliminate the word „quantified“ since no absolute quantification has been made. In fact only a relative quantification based of gas chromatographic area was reported and only in the case of the HS-SPME-GC-MS analysis.

Line 85: delete the word „also“ no other analysis has been presented yet

Table 1: Retention Index should be reported without decimal places since in the most used databases (i.e. NIST) it is reported as an integer number.

 line 94: What do you mean with "were calculated"? 

Line 106: “In each region, the highest concentration of substances is 1-menthol and d-Camphor.” Please rephrase, for example (if I understand well the sense of you sentence): 1-Mentol and d-camphor have the highest concentration in each of the six region under investigation. Furthermore the nouns of the molecules have to be written without capital letter (often in the manuscript I found them in capital letter for example line 241, 242, 244…)

Line 140: „Surprisingly, cubebene were the volatile components only found in A and Z“ please correct „cubebene was“.

Line 172: please correct: „He X. et al“

Line 267-270: „HS-GC-IMS is combined with HS-SPME-GC-MS technology to realize the quick identification and comprehensive characterization of VOCs in SS. By using the well-established HS-GC-IMS technique, 40 VOCs were discovered in SS. On the other hand, 42 VOCs were identified in SS, by the established HS-GC-IMS analysis method.“

Probably there is a mistake I think that you mean … By using the well-established HS-SPME-GC-MS technique, 40 VOCs were discovered in SS…

Line 351: were pre-equilibrated at 60°C

Line 352: please rephrase like for example: „Then, the fiber was immediately insert fiber in the GC-MS injection port where the 230°C hot stripping lasts for 5 min.“

Line 361: correct 110 â—¦C

Author Response

Response to Reviewer 5 Comments

Point 1: Line 22: please eliminate the word “quantified“ since no absolute quantification has been made. In fact only a relative quantification based of gas chromatographic area was reported and only in the case of the HS-SPME-GC-MS analysis.

Response 1: We appreciate the reviewer and agree on this point. In the revision process, we have eliminated the word “quantified“.

Point 2: Line 85: delete the word “also“ no other analysis has been presented yet

Response 2: We appreciate the reviewer and agree on this point. In the revision process, we have eliminated the word “also “.

Point 3: Table 1: Retention Index should be reported without decimal places since in the most used databases (i.e. NIST) it is reported as an integer number.

Response 3: Table 1 shows the data of HS-GC-IMS,VOCs were identified based on the IMS database of HS-GC-IMS Library Search application software rather than the NIST library.

Point 4: line 94: What do you mean with "were calculated"?

Response 4: We appreciate the reviewer and agree on this point. In the revision process, we have  replaced "calculated" with “determined”.

Point 5: Line 106: “In each region, the highest concentration of substances is 1-menthol and d-Camphor.” Please rephrase, for example (if I understand well the sense of you sentence): 1-Mentol and d-camphor have the highest concentration in each of the six region under investigation. Furthermore the nouns of the molecules have to be written without capital letter (often in the manuscript I found them in capital letter for example line 241, 242, 244…)

Response 5: Thank you for your suggestion. Your suggestions are of great help to us. Based on your suggestion, we have modified this sentence. And carefully examined the nouns of the molecules, modifying the capital letters.

Point 6: Line 140: “Surprisingly, cubebene were the volatile components only found in A and Z“ please correct “cubebene was“.

Response 6: Thank you for your suggestion. Your suggestions are of great help to us. Based on your suggestion, we have modified this sentence.

Point 7: Line 172: please correct: “He X. et al“

Response 7: Thank you for your suggestion. Your suggestions are of great help to us. Based on your suggestion, we have modified this sentence.

Point 8: Line 267-270: „HS-GC-IMS is combined with HS-SPME-GC-MS technology to realize the quick identification and comprehensive characterization of VOCs in SS. By using the well-established HS-GC-IMS technique, 40 VOCs were discovered in SS. On the other hand, 42 VOCs were identified in SS, by the established HS-GC-IMS analysis method.“Probably there is a mistake I think that you mean … By using the well-established HS-SPME-GC-MS technique, 40 VOCs were discovered in SS…

Response 8: Thank you for your suggestion. Based on your suggestion, we have modified this sentence. By using the well-established HS-GC-IMS technique, 40 VOCs were discovered in SS. On the other hand, 42 VOCs were identified in SS, by the established HS-SPME-GC-MS analysis method.

Point 9: Line 351: were pre-equilibrated at 60°C

Response 9: We appreciate the reviewer and agree on this point. In the revision process, we have  corrected it.

Point 10: Line 352: please rephrase like for example: “Then, the fiber was immediately insert fiber in the GC-MS injection port where the 230°C hot stripping lasts for 5 min.“

Response 10: We appreciate the reviewer and agree on this point. In the revision process, we have  

corrected it.

Point 11: Line 361: correct 110 â—¦C

Response 11: We appreciate the reviewer and agree on this point. In the revision process, we have  corrected it.

Round 2

Reviewer 1 Report

The authors have respond positively to the most of my comments.